# Prevalence of High-Risk β-Lactam Resistance Genes in Family Livestock Farms in Danjiangkou Reservoir Basin, Central China

**DOI:** 10.3390/ijerph19106036

**Published:** 2022-05-16

**Authors:** Fengxia Yang, Zulin Zhang, Zijun Li, Bingjun Han, Keqiang Zhang, Peng Yang, Yongzhen Ding

**Affiliations:** 1Agro-Environmental Protection Institute, Ministry of Agriculture and Rural Affairs, Tianjin 300191, China; yangfengxiacomeon@163.com (F.Y.); 82101205243@caas.cn (Z.L.); 82101192127@caas.cn (B.H.); yangpeng02@caas.cn (P.Y.); 2China-UK Agro-Environmental Pollution Prevention and Control Joint Research Centre, Tianjin 300191, China; zulin.zhang@hutton.ac.uk; 3The James Hutton Institute, Aberdeen AB15 8QH, UK

**Keywords:** family livestock farms, β-lactam resistance genes, livestock and poultry manure, receiving environment

## Abstract

The propagation of antibiotic resistance genes (ARGs) from domestic livestock manure is an unnegligible important environmental problem. There is an increasing need to understand the role of domestic livestock manure in causing antibiotic resistance in the environment to minimize risks to human health. Here, we targeted β-lactam resistance genes (*bla* genes), primarily discovered in clinical settings, to compare the high-risk ARG profile and their main spreading vectors of 26 family livestock farms in China and analyze the effects of domestic livestock manure on their receiving farmland environments. Results showed that the high-risk *bla* genes and their spreading carriers were widely prevalent in livestock and poultry manure from family farms. The *bla*_ampC_ gene encoding extended-spectrum AmpC β-lactamases, as well as its corresponding spreading carrier (class-1 integron), had the highest occurrence level. The *bla* gene abundance in family chicken farms was higher than that in family swine and cattle farms, while the *bla* gene contamination in the feces of laying hens or beef cattle was worse than that in corresponding broiler chickens or dairy cattle. Notably, the application from domestic livestock manure led to substantial emission of *bla* genes, which significantly increased the abundance of high-risk resistance genes in farmland soil by 12–46 times. This study demonstrated the prevalence and severity of high-risk resistance genes in domestic livestock and poultry manure; meanwhile, the discharge of *bla* genes also highlighted the need to mitigate the persistence and spread of these elevated high-risk genes in agricultural systems.

## 1. Introduction

Antibiotic resistance genes (ARGs), especially the high-risk resistance genes such as extended-spectrum AmpC β-lactamases resistance genes, have become one of the major challenges for human beings [1]. The main mechanism that bacteria is resistant to β-lactam antibiotics is the production of β-lactamases. With the application of β-lactam antibiotics in clinical medicine and livestock industry, various types of β-lactamases including extended-spectrum β-lactamase (ESBL), extended-spectrum AmpC (ESAC), and carbapenemases are emerging [2]. The bacteria that can produce β-lactamases are not only resistant to a series of β-lactam antibiotics, such as penicillin and cephalosporin [3,4,5], but also to non-β-lactam antibiotics such as macrolide, sulfamethoxazole, and quinolones [6,7].

With the abuse of β-lactam antibiotics, more and more β-Lactam resistance genes (*bla* genes) have been detected in animal-derived microorganisms, especially since livestock farms have become another important place where *bla* genes and multidrug-resistant bacteria exist [8]. The *bla* gene-positive *Escherichia coli* (*E. coli*) strains were detected in 15 out of 19 pig farms in Denmark [9]. German scholars once isolated *E. coli* carrying the *bla* gene from the manure in 26 swine farms [10,11]. The *bla*_IMP27_ gene, a special variant of the *bla* gene, was found on a swine farm in the United States [12]. The high prevalence of the high-risk *bla* gene was also found in a previous investigation of large-scale dairy farms and intensive swine farms in China [13,14]. Until now, the *bla* genes have been reported in more than 30 countries, implying a widespread distribution of *bla* genes all over the world [15,16].

Current research mainly focuses on intensive livestock farms, the distribution of *bla* genes in domestic livestock manure and its surrounding farmland environment is unclear, and relevant data are still lacking. The family breeding model is a major mode of breeding in rural areas of China. According to the statistics provided by the Ministry of Agriculture, rural family farms account for more than 58% of the total livestock farms in China [17]. There are many problems in domestic farms in China, such as a large number of family breeding farms, scattered breeding sites, non-standard use of antibiotics, difficulties in waste recycling, and lack of follow-up treatment facilities [18,19,20]. Random discharge of livestock and poultry manure carrying ARGs may bring great pressure to the rural environment. Therefore, analyzing the pollution of ARGs in livestock and poultry manure from family farms and assessing its potential risk to the environment will help to improve the awareness of antibiotic resistance gene contamination in the livestock industry, as well as the comprehensive pollution control of ARG in the environment. 

In this study, we selected 26 domestic livestock and poultry farms (i.e., 8 swine farms, 8 cattle farms, and 10 chicken farms) in the Danjiangkou Reservoir Basin as target farms to investigate the distribution patterns of 13 high-risk *bla* gene subtypes and the genetic markers of their main mobile genetic elements (MGEs) including *intI*1 (class-1 Integrase), *intI*2 (class-2 Integrase), and *tra*A (a genetic marker of conjugate plasmid) by real-time qPCR assays. Understanding the occurrence patterns of the high-risk antibiotic resistance determinants in the domestic livestock farming environment and their contribution to antibiotic resistance to the agro-ecological system will lead to enrich our understanding of antibiotic resistance genes and inform future environmental risk assessment. Therefore, the objectives of the present study were (1) to clarify the occurrence, diversity, and broad distribution of high-risk resistance genes in domestic swine, chicken, and cattle farms and (2) to reveal the effect of domestic livestock and poultry manure on antibiotic resistance in farmland soil.

## 2. Materials and Methods

### 2.1. Experimental Materials

The experimental materials used in this study were all collected from 26 family livestock and poultry farms in 10 villages and towns, such as Xijiadian, Shigu, and Yunyang, in Danjiangkou Reservoir Basin. These family livestock farms, including 8 swine farms, 8 cattle farms (4 beef cattle farms and 4 dairy cattle farms), and 10 chicken farms (5 broiler farms and 5 laying hen farms), have been returning the manure to the field for more than 2 years. In terms of the size of these target family farms, cattle farms ranged from 5 to 15 cattle, the pig amount on hand was 20–50 heads, and chicken farms ranged from 500 to 1500 chickens. The collected samples included livestock manure, breeding wastewater, and manured soil. Among them, there were 42 manure samples (4 beef cow manure, 4 dairy cow manure, 8 sow manure, 8 piglet manure, 8 fattening pig manure, 5 broiler chicken manure, and 5 layer chicken manure), 16 wastewater samples (8 piggery wastewater and 8 cowshed wastewater), and 52 soil samples (26 fertilized soil samples and 26 control soil samples). The manure samples were daily fresh manure, and the wastewater samples were from the raw wastewater pool (as influent) and the outlet of final processing unit (as effluent). Soil samples were the farmland soil with the application of manures, and the control soil was the farmland soil without the application of manures or little human activities. All the samples were frozen and conserved at −20 °C. The details of the distribution of farms in towns and villages along Danjiangkou Reservoir are shown in Figure 1, which was generated using ArcGIS 10.2 software (Esri Inc., Redland, CA, USA).

### 2.2. Sample Collection and Pretreatment

Manure samples were collected at three or five points randomly selected from each dunghill, depending on the size of the manure pile. At each point, the subsamples were taken every 20 cm from the bottom to the top, 200 g each time, and then all the separate subsamples from the same farm were fully mixed into one sample. Soil samples were taken from 0–15 cm of the surface layer by plum blossom sampling method according to the “soil monitoring technical specifications” (HJ/T 166-2004) [21] and then mixed into one sample using the quartering method [22,23]. The manure and soil samples were freeze-dried, sieved, and then stored at −20 °C for the follow-up extraction of total genomic DNA. We also collected piggery wastewater and cattle farm wastewater. The wastewater samples were collected three times randomly using sterile containers from the raw wastewater pool (as influent) and the outlet of the final wastewater processing unit (as effluent), and then, the three repeats of each sampling site from the same farms were mixed as one sample. Every wastewater sample was all filtered with a 0.22 μm millipore membrane. After filtering, the membrane was cut with high-pressure sterilized scissors and put into Lying Matrix E Tube for DNA extraction. 

### 2.3. DNA Extraction and Qualitative PCR Detection

A total of 0.5 g of the manure and soil samples was weighed and used to extract genomic DNA using the Fast DNA SPIN Kit for soil (MP Bio-medicals, LLC, Santa Ana, CA, USA). The above methods were used to extract DNA from the pretreatment wastewater membrane samples. After extraction, the concentration and purity of the DNA solution samples were determined with a microprotein nucleic acid analyzer (nanovue plus, UK BY). Then, each DNA sample went through the traditional quantitative PCR reaction to detect the occurrence of various *bla* gene subtypes, which was performed with a 25 μL reaction system, including 12.5 μL 2 × EasyTaq PCR SuperMix (TransGen Biotech, Beijing, China), the 0.5 μL of upstream and downstream specific primers (10 μmol·L^−1^), 0.5 μL DNA template (being diluted to the appropriate concentration), and 11 μL ddH_2_O. Blank control was also set according to the requirements for each round of reaction, which used the corresponding volume of ddH_2_O instead of the DNA template. 

The conditions of the PCR were followed in a previous study [14]. In detail, the amplification reaction was first pre-denatured at 95 °C for 5 min, then denatured at 95 °C for 30 s, annealed for 30 s (see Table 1 for annealing temperature), extended at 72 °C for 30–60 s, which cycle was performed for 35 times, and finally extended at 72 °C for 7 min. The PCR amplification results were tested by impulsing a 5 μL PCR product in 1 × TAE electrophoresis buffer with 1–1.5% agarose gel at 100 mV for 25 min. The DNA band on the gel was observed with an ultraviolet imager, and the existence of the target gene was determined based on the known molecular band size (DNA marker) of standard DNA. Meanwhile, the positive PCR products were validated by sequencing, and their specificity was tested by Genbank’s Blast program to ensure the accuracy and reliability of the specificity experiment data.

### 2.4. Real-Time qPCR Detection of High-Risk bla Genes

To determine the occurrence abundance of *bla* genes and MGEs in each sample, specific primers of different gene subtypes were selected and amplified by real-time qPCR. The real-time qPCR reaction was carried out in a 7500 Real-Time PCR system (Applied Biosystems). The bacterial 16S rRNA gene was also quantitatively analyzed to characterize and evaluate the total bacterial abundance. The qPCR reaction was performed using a 20 μL reaction system, including 10 μL SYBR^®^ Premix Ex Taq TM II (Tli RNase H Plus, Takara), 0.4 μL of 10 μM upstream and downstream primers, 0.4 μL ROX reference dye, 6.8 μL ddH_2_O, and 2 μL DNA template or standard plasmid DNA. The real-time qPCR conditions were as follows: 95 °C for 30 s followed by 40 cycles at 95 °C for 15 s and 60 °C for 35 s. Melting curve analysis was performed at 60–95 °C and 1 °C/read to ensure the amplification specificity and data accuracy. As previously described, the calibration standard curve pairs for each target gene were generated as positive controls [32]. Negative control (replacing DNA template with ddH_2_O) was set in each run. At the same time, in order to ensure the accuracy of the experiment results, three replicates were set for each sample to detect the copy number of the target gene.

### 2.5. Data Analysis

The abundance, average value, and standard difference of the target genes were calculated by Microsoft Excel 2013. The absolute abundance of ARGs/MGEs and 16S rRNA genes (copies/g DW or mL, copies of target genes per g of dry manure or 1 mL of wastewater) was mapped using OriginPro 8.6 (OriginLab Corporation, Northampton, MA, USA). The SPSS 22.0 software was used for the ANOVA analysis of data, and *p* < 0.05 was taken as the significant difference level. This study also utilized the Pearson correlation coefficient to evaluate the relationship between ARGs abundance and MGEs abundance.

## 3. Results and Discussion 

### 3.1. Occurrence Characteristics of High-Risk Resistance Genes in Livestock and Poultry Waste on Domestic Farms

The results showed that *bla*_ampC_, *bla*_TEM-1_, and *bla*_OXA-1_ genes were common in the livestock and poultry manure from the 26 family farms that combine breeding and planting, which can also be detected in breeding wastewater with detection frequencies of 100%. However, there were some differences in the occurrence of other subtypes of *bla* genes in various livestock and poultry waste (Table 2). In particular, the detection rates of *bla*_NDM_ and *bla*_GES-1_ genes in family cattle waste (62.5% and 81.3%) were higher than those in family swine waste (50.0% and 53.1%). In swine waste samples from family farms, *bla*_CMY-2_, *bla*_SHV_, *bla*_DHA_, and *bla*_KPC-2_ genes were detected at a rate of 12.5%, 21.8%, 9.3%, and 25.0%, respectively, while *bla*_IMP-1_, *bla*_OXA-48_, *bla*_VIM-2_ and *bla*_SPM-1_ subtypes were not detected. In family cattle farm, *bla*_CMY-2_, *bla*_SHV_, *bla*_SPM-1_, *bla*_VIM-2_, and *bla*_DHA_ were detected at a rate of 25.0%, 12.5%, 6.3%, 18.7%, and 6.3% respectively, while *bla*_KPC-2_, *bla*_OXA-48_, and *bla*_IMP-1_ were not detected. Similar to family swine farms and cattle farms, *bla*_NDM_ and *bla*_GES-1_ were detected at a rate of 60% and 40% in chicken waste, while other subtypes of *bla* genes were lower than 30%. The above results revealed that *bla*_ampC_, *bla*_TEM-1_, and *bla*_OXA-1_ were common genes among the tested *bla* genes in livestock and poultry waste on family farms, implying a high prevalence of these gene subtypes and their host bacteria in the corresponding farms.

In order to determine the occurrence level of extended-spectrum ARGs in livestock manure from family farms, the *bla* genes with high detection frequency were further quantified using RT-qPCR. On the whole, the contamination pattern of the *bla* gene in solid manure from chicken farms, pig farms, and cattle farms was: *bla*_ampC_ > *bla*_TEM-1_/*bla*_OXA-1_ > *bla*_GES-1_/*bla*_NDM_ (Figure 2). Among them, the *bla*_ampC_, *bla*_TEM-1_, and *bla*_OXA-1_ were dominant genes with higher abundance, reaching 10^6^~10^12^ copies/g dry weight (DW), 10^5^~10^11^ copies/g DW, and 10^4^~10^9^ copies/g DW, respectively. Although *bla*_NDM_ and *bla*_GES-1_ were common in manure, their contamination levels were as low as 10^4^~10^5^, 10^4^~10^7^, and 10^4^~10^9^ copies/g DW, respectively. In the wastewater from family farms, the *bla*_ampC_ level was higher than that of other *bla* genes, and its abundance was within the range of 10^4^–10^7^ copies/mL, followed by *bla*_TEM-1_ and *bla*_OXA-1_, of which the abundance was 10^2^~10^7^ copies/mL. In contrast, the levels of *bla*_GES-1_ and *bla*_NDM_ were 1~2 orders of magnitude lower, and their abundance was in the range of 10^2^~10^5^ copies/mL (Figure 3).

The comprehensive results of detection frequencies and occurrence levels indicated that ESBL gene (*bla*_TEM-1_ and *bla*_OXA-1_), broad-spectrum ampC β-lactamase gene (*bla*_ampC_), and carbapenemase gene (*bla*_NDM_ and *bla*_GES-1_) were the dominant bacterial resistance determinants to broad-spectrum β-lactam antibiotics in domestic livestock and poultry manure. Notably, the abundance ranges of most *bla* genes on family breeding farms were higher than those on large-scale farms; for example, the abundance of *bla* genes in large-scale farms in Tianjin ranged from 10^3^ to 10^9^ copies/g DW [13], while the maximum abundance of corresponding high-risk genes in this study reached 10^10^ copies/g DW. This result revealed the severity of contamination of high-risk β-lactam resistance genes contaminant in the family farming environment.

### 3.2. Occurrence Level and Variations of High-Risk Resistance Genes in Manure from Different Animals on Family Livestock Farms

Although the occurrence trend in abundances of the *bla* genes in the three kinds of livestock manure was as follows: *bla*_ampC_ > *bla*_TEM-1_> *bla*_OXA-1_> *bla*_NDM_> *bla*_GES-1_, which was similar to the results previously reported on the large-scale farms [14,35], the contamination level of these *bla* gene subtypes in the manure from different animals was different. The results showed that the contamination levels of most *bla* genes in family chicken manure (10^4^~10^12^ copies/g DW) were significantly higher than that in swine manure (10^3^~10^10^ copies/g DW) and cattle manure (10^4^~10^9^ copies g DW) (*p* < 0.05). In particular, *bla*_ampC_ gene encoding AmpC β-lactamase had an average abundance of (6.25 ± 0.07) × 10^11^ copies/g DW, which was much higher than that of swine farms and cattle farms, at the level of (2.34 ± 0.12) × 10^10^ copies/g DW and (2.54 ± 0.16) × 10^9^ copies/g DW, respectively. In domestic chicken manure, the contamination level of *bla*_GES-1_ was significantly lower than that of other *bla* genes (including the *bla*_NDM_ gene) (*p* < 0.05), with the average abundance of only 10^4^ copies/g DW, which was 3~7 orders of magnitude lower than that of other genes. However, in domestic cattle and swine manure, the contamination of *bla*_NDM_ and *bla*_GES-1_ was basically at the same level. The key resistance gene *bla*_NDM_ carried by superbugs was highest in domestic chicken manure (10^4^~10^8^ copies/g DW), while its occurrence levels in swine manure and cattle manure were also as high as 10^3^~10^7^ copies/16S copies. The average abundances of *bla*_TEM-1_ and *bla*_OXA-1_ were (7.92 ± 0.21) × 10^10^ and (1.01 ± 0.36) × 10^9^ copies/g DW (for chicken farms), (1.53 ± 0.21) × 10^9^ and (1.30 ± 0.16) × 10^8^ copies/g DW (for swine farms), and (1.05 ± 0.11) × 10^8^ and (4.44 ± 0.21) × 10^7^ copies/g DW (for cattle farms), respectively.

The occurrence differences of *bla* genes in manures from different animals were closely related to metabolic capacity and breeding methods. Taking family chicken farms as an example, β-lactam antibiotics were often used for infectious diseases in chickens, leading to higher levels of *bla*_OXA-1_, *bla*_TEM_, and *bla*_ampC_ genes detected in chicken manure, which made the pathogenic bacteria in chickens resistant to β-Lactam antibiotics [36,37]. Wang et al. [38] found that the antibiotic residue in domestic chicken manure was much higher than that in family swine and cattle manure, also indicating that the antibiotic dose fed to chickens was higher than cattle. Additionally, compared with swine and cattle (mammals), chickens have a poor digestive function, resulting in residual antibiotics in the chicken gut, further providing the superior conditions for the production and persistence of β-Lactam-resistant bacteria [39]. Previous studies also reported that the level of common ARGs in poultry (chicken and duck) manure was higher than that in swine and cattle manure [40,41]. 

There were also differences in the abundance of high-risk *bla* genes in livestock and poultry manure at different growth stages. The abundances of *bla*_ampC_, *bla*_TEM-1_, and *bla*_OXA-1_ in sow manure were significantly about 1~3 orders of magnitude higher than those of finishing pigs and piglets (*p* < 0.05), mainly due to the differences in the use of antibiotics (such as dose and frequency) and dietary factors (such as feed ratio, dose, and frequency) in swine at different growth stages [42,43].

The distribution of *bla* genes also varied in the wastewater from different livestock farms. For instance, the abundances of *bla*_ampC_, *bla*_OXA-1_, and *bla*_TEM-1_ in wastewater on dairy cattle farms were higher than those in beef cattle farms (Figure 4), but this result was not found in the solid manure of these two kinds of cattle species. Meanwhile, the *bla*_NDM_ gene was detected in all family laying hen manures but not in family broiler manures (Figure 2). We speculated that there may be two reasons for this: (1) different types of antibiotics are used in the breeding and management of layers and broilers; (2) the corresponding host bacteria of the *bla* genes had different living conditions in livestock intestines [44,45]. However, for family beef cattle and dairy cattle, the contamination difference of *bla* genes may be due to different growth and reproduction conditions for bacterial flora on various types of cattle farms [13,46], but overall, there was no significant difference in the *bla* contamination levels between the two types of cattle manures (*p* > 0.05).

### 3.3. High Prevalence and Variation of bla Gene-Related Spreading Vectors in Livestock Manure on Family Breeding Farms 

Mobile genetic elements are the spreading vectors and carriers of ARGs in the environment, and bacteria can acquire ARGs through multiple pathways [47]. Among them, MGEs such as class-1 integron, class-2 integron, and conjugative plasmids play an important role in capturing and transferring ARGs (including high-risk *bla* genes), which is also the reason for the widespread transmission of ARGs all over the world. At present, there are few studies on MGEs related to *bla* genes in family livestock and poultry manure, and the existence, abundance, and differences of mobile genetic factors in different livestock breeding environments are not clear. Given these questions, we further conducted a comprehensive investigation on MGEs in livestock and poultry waste of 26 family farms. Results showed that *tra*A (conjugative plasmid), *intI*1 (class-1 integrase), and *intI*2 (class-2 integrase) were detected in all manure samples in family chicken farms, with a detection rate of 100% (Figure 5). The detection rates of *int**I*1 and *int**I*2 in manures and wastewater samples from swine farms and cattle farms were all 100%, while the detection frequencies of *tra*A were 62.5~87.5% and 68.7~87.5%, respectively. The above results demonstrated that conjugated plasmids, class-1 integron, and class-2 integron occurred widely in livestock and poultry manure, which further indicated that the high prevalence of MGEs is closely related to the spread of *bla* genes in family livestock farming environment.

To clarify the occurrence level and distribution pattern of these typical MGEs in family livestock farms, we further analyzed MGEs in manure by real-time qPCR assay. The quantization results showed that the levels of two class integrons were significantly higher than that of conjugated plasmids in the livestock manure on these three types of family farms (Figure 5). Most manure from family swine farms had the highest *intI*1 gene level, with an abundance ranging from 10^6^ to 10^10^ copies/g DW, but the abundance of genetic marker *tra*A for conjugative plasmid was about 2~3 orders of magnitude lower than that of i*nt**I*1, and its abundance level was at 10^4^~10^8^ copies/g DW. The occurrence characteristics of these typical MGEs in family livestock waste were similar to those of large-scale swine manure [48]. Additionally, we found a similar trend in wastewater samples on swine farms; that is, these MGEs in piggery wastewater also showed the distribution pattern of *intI*1 > *intI*2 > *tra*A (Figure 6). It is noteworthy that MGEs occur at a higher level in sow manure than that in piglet and finishing swine manure, which may be one of the reasons for the higher level of ARGs in sow manure [14,35]. Unlike MGEs in domestic swine manure, the *intI*1 and *intI*2 occurred at the same level in domestic cattle manure, both at 10^3^~10^9^ copies/g DW, but the *tra*A level was at 10^4^~10^8^ copies/g DW. However, the distribution pattern of these MGEs in cattle farm wastewater was basically consistent with that in piggery wastewater, showing the occurrence trend of *intI*1 > *intI*2 > *tra*A. Compared with swine farms and cattle farms, MGEs in chicken manure were more common and abundant, especially with the abundance of conjugative plasmids (*tra*A) as high as 10^8^~10^11^ copies/g DW (Figure 6). This further increased the propagation and dissemination risk of *bla* genes and improved their contamination level in chicken farm manure, which may be one of the main reasons for the higher abundance of *bla* genes in family chicken manure.

The migration of *bla* genes in the environment mainly depends on mobile genetic factors such as plasmids and integrons [49,50]. Through correlation analysis, MGEs level was positively correlated with the abundance of *bla* genes (Table 3), indicating that enhancing the elimination of plasmids and other MGEs in manures would alleviate the release of *bla* genes in livestock manure to some extent, so as to curb the spread and proliferation of ARGs.

### 3.4. Effects of Family Livestock Manure Application on Contamination of High-Risk Resistance Genes in Farmland Soil

Returning manure to farmland is an important measure to solve manure contamination and realize the planting and breeding cycle. This study also investigated the farmland soil with the long-term application of family livestock manure. The corresponding results revealed that among all the *bla* genes, the average level of *bla*_ampC_ was the highest in the manured soil, followed by *bla*_OXA-1_ and *bla*_TEM-1_, which were 1~3 orders of magnitude higher than the control soil (Figure 7). Although the levels of *bla*_NDM_ and *bla*_GES-1_ were low (10^3^~10^5^ copies/g DW) among the tested bla genes, they were widely distributed in farmland soil, with the detection frequencies of 65.3% (34/52) and 26.9% (14/52), respectively. The emergence of *bla*_NDM_ in farmland soil further increased the ecological and health risks of *bla* genes because the NDM enzyme encoded by *bla*_NDM_ can confer bacteria resistant to most β-lactam antibiotics and even other non-beta-lactams [6]. In addition, *bla*_TEM-1_ and *bla*_OXA-1_ could encode ESBLs, which are the most important enzymes that lead to the resistance of β-lactam antibiotics to clinical pathogens, and ESBLs can hydrolyze not only penicillin but also cephalosporins [51]. Although *bla* genes in farmland soil were significantly lower than those in livestock manure (about 3~5 orders of magnitude), their abundances were much higher than that in control soil without livestock manure, highlighting their high stability and risks in the soil environment.

The application of different family livestock manure had different effects on the *bla* gene contamination in the farmland soil environment. Compared with the control soil, the total absolute abundance of *bla* genes in the soil applied with domestic chicken manure increased by 46.3 times, while the *bla* genes abundance in the soil applied with family swine manure and cattle manure increased by 22.3 times and 13.4 times, respectively. The influence degree showed the following trend: chicken manure > swine manure > cow manure, which indicated that the reduction of ARGs in chicken manure is the key to controlling ARGs in the farmland environment. Thus, we could remove the *bla* gene by thermophilic composting of chicken waste [52]. Additionally, in agricultural practice, the farmers could reduce the amount and spread the risk of ARGs by the biochar addition to topsoil or combined application with manure [53]. Moreover, the above results were basically consistent with the occurrence pattern of ARGs in different livestock and poultry manure, showing that the difference in *bla* genes of livestock and poultry manure in family farms was the main reason for the difference in high-risk resistance gene contamination in farmland soil applied with various manures. Additionally, the diversity of exogenous ARG host bacteria in soils with various manures and differences in their survival ability in the farmland of different regions also leads to different persistence of ARGs in the environment [54]. The detection rate and abundance of integrons and conjugative plasmids in manured farmland soil were significantly higher than those in control soil (*p* < 0.05), and their distribution pattern was consistent with that in manure, showing a trend of *intI*1 > *intI*2 > *tra*A (Figure 8). The presence of MGEs increased the dissemination risk of *bla* genes in the soil to some extent [55].

The influences of different livestock and poultry manures on MGEs in farmland soil were also different, and the influence degree of MGEs was basically consistent with that of *bla* genes (Figure 8). Compared with the control soil, the increase in MGEs abundance in the soil applied with chicken manure (43.7 times) was significantly higher than that with swine manure (23.5 times) and cow manure (12.8 times). In conclusion, except that carbapenem-resistant genes were rarely detected or existed in low abundances in the soil applied with these three different manures, other extended-spectrum *bla* genes generally existed at high abundances in fertilized soils, which can fully prove that livestock and poultry manure from family farms is an important contamination source of high-risk *bla* genes in farmland soil. Therefore, it is imperative to use effective manure treatment techniques to control the spread of antibiotic resistance genes, especially those high-risk multidrug-resistance determinants in family livestock and poultry manures.

## 4. Conclusions

This study demonstrated the high prevalence and diversity of high-risk β-Lactamase resistance genes in domestic livestock and poultry manure in Danjiangkou Reservoir Basin. We determined that *bla*_ampC_, *bla*_TEM-1_, and *bla*_OXA-1_ were the dominant resistance factors causing bacteria in livestock manure to be resistant to β-lactam antibiotics. There were differences in the contamination degree of *bla* genes in different family livestock and poultry manure; the contamination of manure from family chicken farms was more serious than that from family swine and cattle farms. Additionally, different growth and development stages and different varieties also had an effect on *bla* genes contamination in manure. Moreover, the class-1 and class-2 integrons and conjugative plasmids, which were closely related to the spread of ARGs, were widely found in domestic livestock manure and basically consistent with the distribution pattern of *bla* genes, but the occurrence pattern in different livestock and poultry manures varied. Even more troubling, returning the domestic livestock and poultry manure to farmlands not only significantly increased the abundance of *bla* genes in soil but also increased the risk of its spreading in the agricultural system, revealing that domestic livestock and poultry manure is a non-ignored important pollutant resource of ARGs and multidrug-resistant determinants in the agricultural ecosystem.

## Figures and Tables

**Figure 1 ijerph-19-06036-f001:**
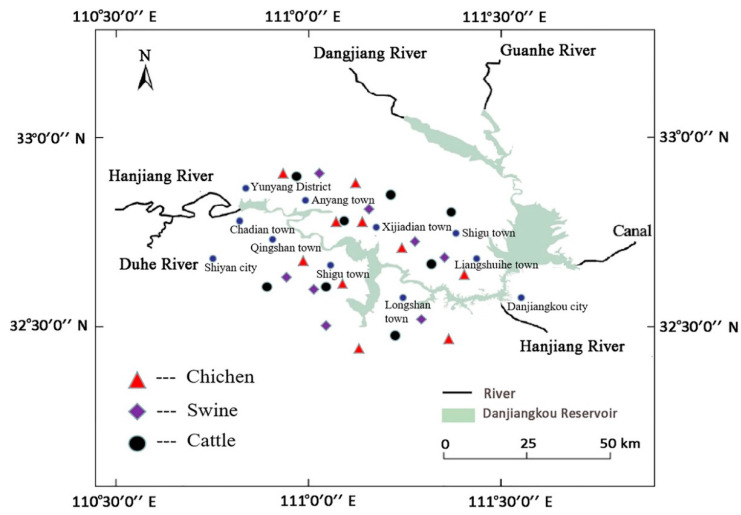
Distribution map of the target family farms in Danjiangkou Reservoir Basin.

**Figure 2 ijerph-19-06036-f002:**
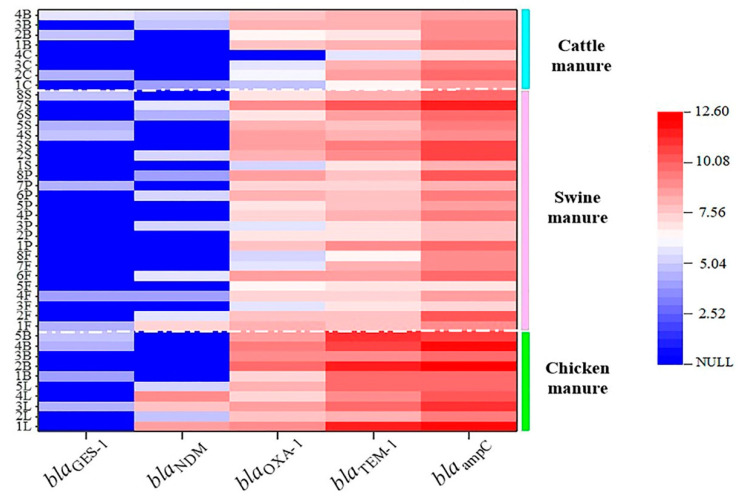
Heat maps of the abundance of *bla* genes in the family livestock manure. Abbreviations (from top to bottom): B—beef cow manure; C—dairy cow manure; S—sow manure; P—piglet manure; F—fattening pig manure; B—broiler chicken manure; L—layer chicken manure; 1–8—serial number of farm; NULL—no detected.

**Figure 3 ijerph-19-06036-f003:**
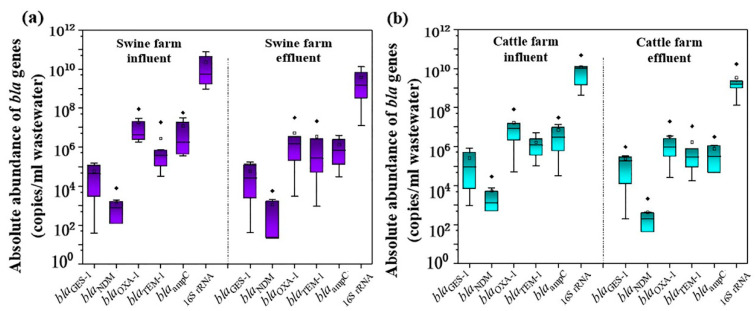
Absolute abundance of *bla* genes in livestock wastewater from family swine farms (**a**) and family cattle farms (**b**). Solid points represent extreme outliers, that is, outliers more than three times the quartile distance; Hollow points represent milder outliers, that is, outliers between 1.5 and 3 quartiles.

**Figure 4 ijerph-19-06036-f004:**
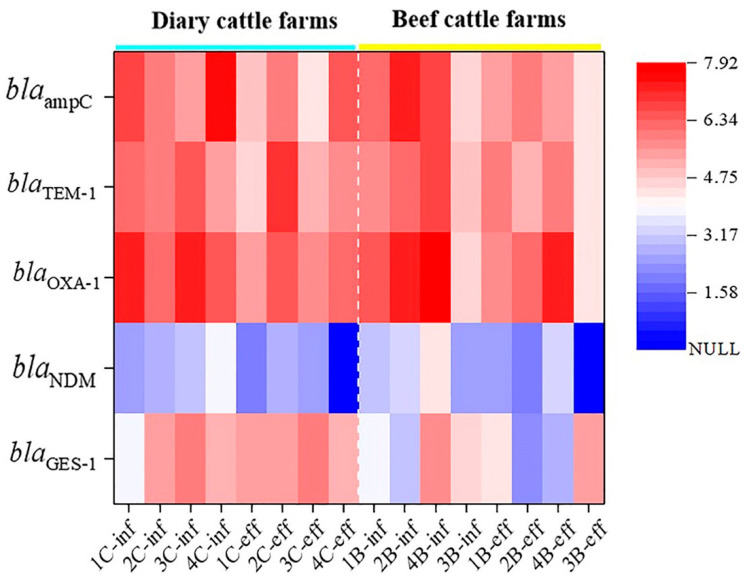
Heat maps of the absolute abundance of *bla* genes in raw influent (-inf) and final effluent (-eff) from dairy cattle and beef cattle farms.

**Figure 5 ijerph-19-06036-f005:**
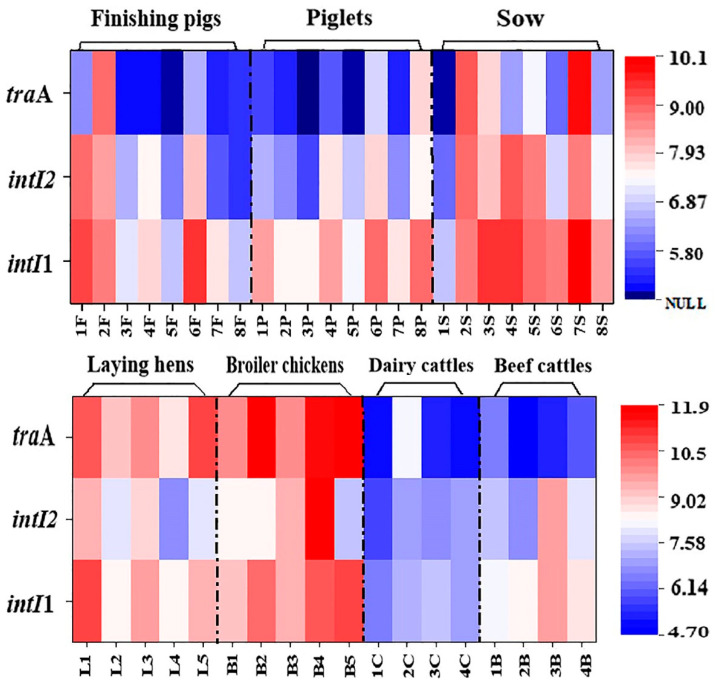
The concentration of MGEs in livestock manure from family farms.

**Figure 6 ijerph-19-06036-f006:**
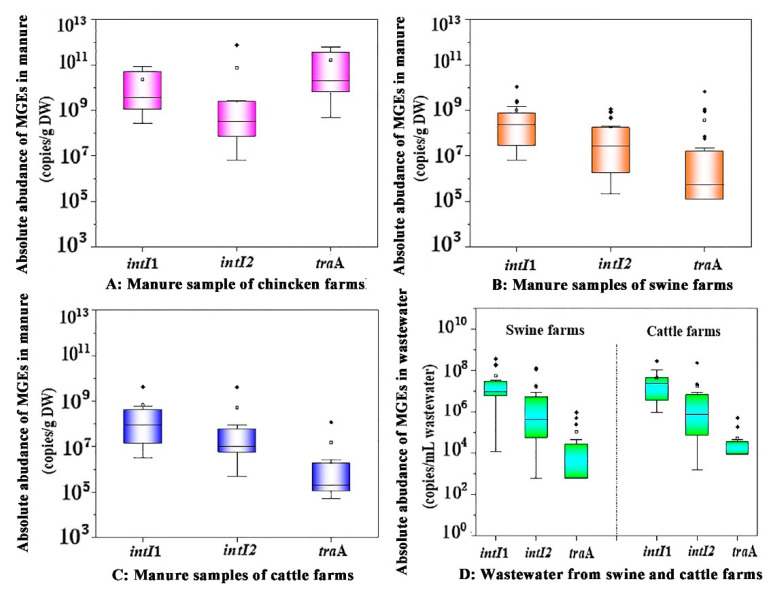
The absolute abundance of MGEs in livestock feces (**A**–**C**) and wastewater (**D**) from family farms. Solid points represent extreme outliers, that is, outliers more than three times the quartile distance; Hollow points represent milder outliers, that is, outliers between 1.5 and 3 quartiles.

**Figure 7 ijerph-19-06036-f007:**
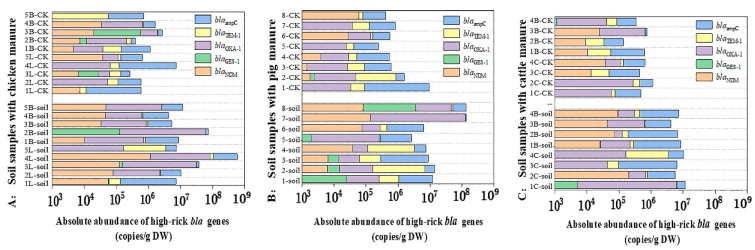
The absolute abundance of *bla* genes in soil samples applied with livestock manure of family farms.

**Figure 8 ijerph-19-06036-f008:**
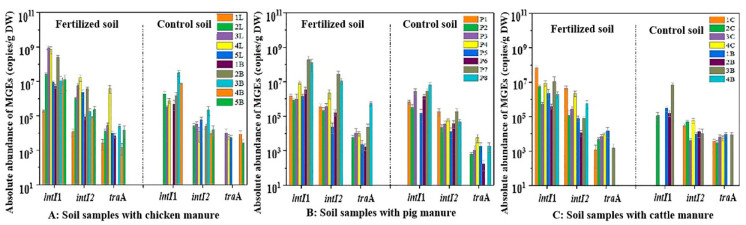
The absolute abundance of MGEs in soil samples applied with livestock manure of family farms.

**Table 1 ijerph-19-06036-t001:** Primer information used in this study.

Target Genes	Primer Sequences (5′-3′)	Size (bp)	References
*bla* _OXA-1_	F-TATCTACAGCAGCGCCAGTG	199	[24]
	R-CGCATCAAATGCCATAAGTG		
*bla* _ampC_	F-CCTCTTGCTCCACATTTGCT	189	[24]
	R-ACAACGTTTGCTGTGTGACG		
*bla* _TEM-1_	F-CATTTTCGTGTCGCCCTTAT	167	[24]
	R-GGGCGAAAACTCTCAAGGAT		
*bla* _GES-1_	F-ATGGCACGTACTGTGGCTAA	287	[5]
	R-TGACCGACAGAGGCAACTAAT		
*bla* _IMP-1_	F-GGAATAGAGTGGCTTAAYTCTC	232	[25]
	R-GGTTTAAYAAAACAACCACC		
*bla* _VIM-2_	F-GTTTGGTCGCATATCGCAAC	382	[25]
	R-AATGCGCAGCACCAGGATAG		
*bla* _KPC-2_	F-ATGTCACTGTATCGCCGTCT	893	[26]
	R-TTTTCAGAGCCTTACTGCCC		
*bla* _OXA-48_	F-GCGTGGTTAAGGATGAACAC	438	[25]
	R-CATCAAGTTCAACCCAACCG		
*bla* _DHA_	F-AACTTTCACAGGTGTGCTGGGT	405	[27]
	R-CCGTACGCATACTGGCTTTGC		
*bla* _CMY-2_	F-TGGCCAGAACTGACAGGCAAA	462	[28]
	R-TTTTCCTGAACGTGGCTGGC		
*bla* _SPM-1_	F-AAAATCTGGGTACGCAAACG	271	[29]
	R-ACATTATCCGCTGGAACAGG		
*bla* _SHV_	F-GGGTTATTCTTATTTGTCGC	930	[30]
	R-TTAGCGTTGCCAGTCCTC		
*traA*	F-AAAGAATTCGAAATTGAGGTAACTTATGAATGC	58	[31]
	R-CCCAAGCTTCGTTTTATTTCCTGTCAGAG		
*intI*1	F-GGCTTCGTGATGCCTGCTT	55	[32]
	R-CATTCCTGGCCGTGGTTCT		
*intI*2	F-TTATTGCTGGGATTAGGC	58	[33]
	R-ACGGCTACCCTCTGTTATC		
16S rRNA	F-CGGTGAATACGTTCYCGG	126	[34]
	R-GGWTACCTTGTTACGACTT		

**Table 2 ijerph-19-06036-t002:** Detection rate of *bla* genes in livestock waste from different family farms.

Gene	Swine Waste	Cattle Waste	Chicken Waste
*bla* _OXA-1_	100% (32/32)	100% (16/16)	100% (10/10)
*bla* _ampC_	100% (32/32)	100% (16/16)	100% (10/10)
*bla* _TEM-1_	100% (32/32)	100% (16/16)	100% (10/10)
*bla* _GES-1_	53.1% (17/32)	81.3% (13/16)	40.0% (4/10)
*bla* _NDM_	50.0% (16/32)	62.5% (10/16)	60.0% (6/10)
*Bla* _CMY-2_	12.5% (4/32)	25.0% (4/16)	20.0% (2/10)
*bla* _IMP-1_	0.0% (0/32)	0.0% (0/16)	0.0% (0/10)
*bla* _VIM-2_	0.0% (0/32)	18.7% (3/16)	10.0% (1/10)
*bla* _OXA-48_	0.0% (0/32)	0.0% (0/16)	0.0% (0/10)
*bla* _DHA_	9.3% (3/32)	6.3% (1/16)	10.0% (1/10)
*bla* _SPM-1_	0.0% (0/32)	6.3% (1/16)	10.0% (1/10)
*bla* _SHV_	21.8% (7/32)	12.5% (2/16)	20.0% (2/10)
*bla* _KPC-2_	25.0% (8/32)	0.0% (0/16)	10.0% (1/10)

**Table 3 ijerph-19-06036-t003:** The correlation between high-risk *bla* genes and MGEs.

The *bla* Genes		*bla* _NDM_	*bla* _GES-1_	*bla* _OXA-1_	*bla* _TEM-1_	*bla* _ampC_
Chicken manure	*intI*1	0.139	0.690 *	0.228	0.502	0.213
	*intI*2	0.149	0.129	0.276	0.113	0.187
	*tra*A	0.302	0.479	0.782 **	0.713 *	0.737 *
Swine manure	*intI*1	0.061	0.032	0.864 **	0.946 **	0.946 **
	*intI*2	0.412 *	0.536 **	0.637 **	0.179	0.217
	*tra*A	0.042	0.132	0.675 **	0.975 **	0.980 **
Cattle manure	*intI*1	0.602	0.054	0.978 **	0.036	0.216
	*intI*2	0.523	0.161	0.957 **	0.027	0.182
	*tra*A	0.232	0.114	0.212	0.895 **	0.936 **

* Correlation is significant (*p* < 0.05). ** Correlation is highly significant (*p* < 0.01).

## Data Availability

The data presented in this study are available on request from the corresponding author.

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
