# Peer review of "Prevalence of High-Risk β-Lactam Resistance Genes in Family Livestock Farms in Danjiangkou Reservoir Basin, Central China"

_ijerph, 2022, doi:10.3390/ijerph19106036_

Round 1

Reviewer 1 Report

Well presented sound scientific article but need improvement in some areas: 

  1. Table 1 headings should be translated to English to identify the primers.
  2. Typesetting and typographical errors should be checked such as in line 128 and 129 where the word "determined" was displaced.
  3. Sample size for each sample types appear small but still acceptable due to molecular analysis using 16 primers (15 for antimicrobial genes and 1 for bacteria gene) 

Author Response

Mar 11, 2022

Re: Manuscript Number: ijerph-1696465

Dear editor and the reviewer,

Thank you very much for processing our paper and giving us the opportunity to revise the paper. We are also grateful to the editor and reviewers’ comments for their insightful comments, which improve the quality of this paper. According to the suggestions of the editor and reviewers, we have carefully taken into account of all points and made corresponding improvements to the manuscript.

According to the suggestions of reviewers, we supplemented the sample collection part of the manuscript in more detail, including method of sampling. Meanwhile, we have adjusted the clarity of all the figures and enlarged the font of each figure to make the figures clearer. We also added the more discussion to clarify our results for the readers and further improve our conclusions in the revised manuscript. In addition, we answered the questions raised by the reviewers one by one and contacted a professional teacher to check and modify the grammar of our manuscript.

All comments were addressed item by item through extensive text clarification and provision of additional data and discussion. Attached is a point-by-point response to the reviewers’ comments and a revised paper with highlighted changes to facilitate your evaluation of changes made. We hope that all these changes fulfill the requirements to make the manuscript acceptable for publication in International Journal of Environmental Research and Public Health.

Looking forward to hearing from you soon.

a point-by-point response , Please see the attachment.

Yours truly,

Keqiang Zhang

Reviewer 2 Report

The authors describe the detection of b-lactam resistance genes from domestic livestock farms in China.  It was demonstrated a prevalence of such genes in the samples analyzed.  Overall, I think the manuscript lacks a strong closing argument for their findings.  Their main conclusion is that these ARGs are present in the samples tested, which it is expected.  How significant is the prevalence of these genes?  How the authors propose to solve the problem?  Are these genes active in the manure?  Also, a person fluent in English should revise the manuscript.

Minor Comments:

  • Table 1: some headings are in Chinese
  • Figure 7 and 8 are too small it is difficult to read

Author Response

Mar 11, 2022

Re: Manuscript Number: ijerph-1696465

Dear editor and the reviewer,

Thank you very much for processing our paper and giving us the opportunity to revise the paper. We are also grateful to the editor and reviewers’ comments for their insightful comments, which improve the quality of this paper. According to the suggestions of the editor and reviewers, we have carefully taken into account of all points and made corresponding improvements to the manuscript.

According to the suggestions of reviewers, we supplemented the sample collection part of the manuscript in more detail, including method of sampling. Meanwhile, we have adjusted the clarity of all the figures and enlarged the font of each figure to make the figures clearer. We also added the more discussion to clarify our results for the readers and further improve our conclusions in the revised manuscript. In addition, we answered the questions raised by the reviewers one by one and contacted a professional teacher to check and modify the grammar of our manuscript.

All comments were addressed item by item through extensive text clarification and provision of additional data and discussion. Attached is a point-by-point response to the reviewers’ comments and a revised paper with highlighted changes to facilitate your evaluation of changes made. We hope that all these changes fulfill the requirements to make the manuscript acceptable for publication in International Journal of Environmental Research and Public Health.

Looking forward to hearing from you soon.

A point-by-point response , please see the attachment.

Yours truly,

Keqiang Zhang

Reviewer 3 Report

The manuscript investigated the occurrence, diversity and distribution of high-risk resistance genes in domestic swine, chicken and cattle farms. The manuscript has a piece of valuable information; however, it is not reasonably explained and structured. The standard of English needs to be improved to make the report easier to read and remove unnecessary repeated information. The authors should address the following significant and major points and I will delay my decision until check the restructured version.

  • The objective of the study is to show the diversity and distribution. Thus, the authors need to clarify the principle used to select the village, towns, and farms? And also, how the sample size was calculated?
  • The authors missed reporting discussion section! However, I believe that the results section includes the discussion. I did not know if the journal format accepts merging results and discussion under one section. However, I recommend that authors separate them to give a more deep discussion to clarify their results for the readers.
  • Results, section 3.1, is highly confusing! I suggest designing a table which includes the number of samples collected from each farm and the number of positive genes (%).
  • The quality of all figures is bad, and the data are not clear. Please provide figures with high resolution.
  • Line 323: Table 2 is missing. Please provide the table.

Minor points:

Line 71-78: please rephrase as follows “Understanding the occurrence patterns of the high-risk antibiotic resistance determinants in domestic livestock farming environment and their contribution on antibiotic resistance of agro-ecological system will lead to enrich our understanding of antibiotic resistance genes and inform future environmental risk assessment. Therefore, the objectives of the present study were (1) to clarify the occurrence, diversity and broad distribution of high-risk resistance genes in domestic swine, chicken and cattle farms and (2) to reveal the effect of domestic livestock and poultry manure on antibiotic resistance in farmland soil.”

Line 95: Figure 1, Please add the source or the map or the software used to generate it.

Line 98: The sample was collected every 20 cm? authors could you please clarify how many samples were collected before mixing into one sample?

Line 99-100: quartering method and plum blossom sampling method, please add references for these methods.

Line 103-104: Do the authors means that samples from pig and cattle houses are mixed in one sample? if yes, why?

Line 128: correct the sentence

Line 133: Table 1, please used the English language 

Author Response

(The authors gave the same response as above.)

Round 2

Reviewer 2 Report

I am glad to see my comments were taken in consideration to improve the manuscript.

Author Response

Thank you very much for your valuable comments on our manuscript.

Reviewer 3 Report

This manuscript has been significantly improved over its original version. The authors made substantial changes to the Materials and Methods, as well as the Results. Despite, this reviewer still finds some points not satisfactory that need to be corrected before final approval for publication:

Line 112: delete “respectively”

Line 115: correct “form” to “from”

Line 170: authors indicated in the revised version that results and discussion combined in one section. Please, change the subtitle to “Results and discussion”.

Author Response

Thank you very much for your valuable comments on our manuscript.

A point-by-point response , please see the attachment.
